# Biodistribution of Intratracheal, Intranasal, and Intravenous Injections of Human Mesenchymal Stromal Cell-Derived Extracellular Vesicles in a Mouse Model for Drug Delivery Studies

**DOI:** 10.3390/pharmaceutics15020548

**Published:** 2023-02-07

**Authors:** Anna Maria Tolomeo, Gaia Zuccolotto, Ricardo Malvicini, Giada De Lazzari, Alessandro Penna, Chiara Franco, Federico Caicci, Fabio Magarotto, Santina Quarta, Michela Pozzobon, Antonio Rosato, Maurizio Muraca, Federica Collino

**Affiliations:** 1Department of Cardiac, Thoracic and Vascular Science and Public Health, University of Padova, 35128 Padua, Italy; 2L.i.f.e.L.a.b. Program, Consorzio per la Ricerca Sanitaria (CORIS), Veneto Region, 35128 Padua, Italy; 3Istituto Oncologico Veneto IOV-Istituto di Ricovero e Cura a Carattere Scientifico IRCCS, 35128 Padua, Italy; 4Department of Women’s and Children’s Health, University of Padova, 35128 Padua, Italy; 5Instituto de Medicina Traslacional, Trasplante y Bioingeniería, Universidad Favaloro-Consejo Nacional de Investigaciones Cientìficas y Técnicas, Buenos Aires 1078, Argentina; 6Department of Surgery, Oncology and Gastroenterology, University of Padova, 35128 Padua, Italy; 7Rheumatology Unit, Department of Medicine (DIMED), University of Padova, 35128 Padua, Italy; 8Department of Biology, University of Padova, 35131 Padua, Italy; 9Department of Medicine, University of Padova, 35128 Padua, Italy; 10Fondazione Istituto di Ricerca Pediatrica Città della Speranza, 35127 Padua, Italy; 11Laboratory of Translational Research in Paediatric Nephro-Urology, Department of Clinical Sciences and Community Health, University of Milano, 20122 Milan, Italy; 12Fondazione Ca’ Granda IRCCS Ospedale Maggiore Policlinico, 20122 Milan, Italy

**Keywords:** extracellular vesicles, biodistribution, pharmacokinetics, drug delivery

## Abstract

Mesenchymal stromal cell-derived extracellular vesicles (MSC-EVs) are extensively studied as therapeutic tools. Evaluation of their biodistribution is fundamental to understanding MSC-EVs’ impact on target organs. In our work, MSC-EVs were initially labeled with DiR, a fluorescent lipophilic dye, and administered to BALB/c mice (2.00 × 10^10^ EV/mice) through the following routes: intravenous (IV), intratracheal (IT) and intranasal (IN). DiR-labeled MSC-EVs were monitored immediately after injection, and after 3 and 24 hours (h). Whole-body analysis, 3 h after IV injection, showed an accumulation of MSC-EVs in the mice abdominal region, compared to IT and IN, where EVs mainly localized at the levels of the chest and brain region, respectively. After 24 h, EV-injected mice retained a stronger positivity in the same regions identified after 3 h from injection. The analyses of isolated organs confirmed the accumulation of EVs in the spleen and liver after IV administration. Twenty-four hours after the IT injection of MSC-EVs, a stronger positivity was detected selectively in the isolated lungs, while for IN, the signal was confined to the brain. In conclusion, these results show that local administration of EVs can increase their concentration in selective organs, limiting their systemic biodistribution and possibly the extra-organ effects. Biodistribution studies can help in the selection of the most appropriate way of administration of MSC-EVs for the treatment of different diseases.

## 1. Introduction

Extracellular vesicles (EVs) are heterogeneous small membrane structures originating from plasma membranes and containing molecules from the cells from which they are released. EVs function by transferring biological molecules to target cells that accounts for an active alteration of the cell phenotype and function [1,2], both in physiological and pathological conditions. EVs are present in almost all body fluids and can transmigrate through different tissue barriers (e.g., blood–brain barrier, BBB) reaching distant sites [3,4]. Due to these biological and functional characteristics, EVs have been proposed as a novel therapeutic platform for delivering pharmaceutical ingredients [5]. In support of their application, EVs show high biocompatibility as well as low immunogenicity in different tissues [6,7].

Mesenchymal stem cells (MSCs) play an important role in the regeneration of multiple organs [8]. It has been shown that the MSC-derived EVs (MSC-EVs) partly mediate the MSC therapeutic functions through a paracrine way. Their mechanism of action can be summarized in their capability to directly regenerate different tissues, by protecting from damage-related inflammation events and inducing pro-proliferative and angiogenic programs in different organs [9,10,11,12,13,14]. On the clinical side, MSC-EVs provide important advantages, in respect to their cells of origin, such as their reduced immunogenicity and their efficient homing and uptake by different organs [15,16]. Hence, MSC-EVs can be defined as an alternative regenerative tool to cell-based therapy. Multiple factors can affect MSC-EVs biological activity and, therefore, their translation into medical treatments. The originating cells and their surrounding microenvironment can regulate EVs function by accounting for their membrane and intravesicular composition. For instance, the injection of human-adipose-derived (AD)MSC-derived EVs, in a rat myocardial-infarction model, leads to a stronger protection of cardiomyocytes from apoptosis and to an increase in angiogenesis, compared with one prompted by the bone-marrow (BM)MSC-derived EVs [17]. By contrast, in murine acute-kidney and lung-injury models, inducible pluripotent stem cells (iPSCs) as well as BMMSC-derived EVs were shown to better induce M2 macrophage polarization, compared to that induced by ADMSC-EVs [18,19,20]. The routes of administration may also influence their tissue distribution and consequently their pharmacokinetic profiles, as well as the biological activity [21]. By comparing the intraarterial- and intravenous (IV)-injection strategies of umbilical cord MSC-EVs, Li et al. [22] demonstrated that the less-invasive IV approach resulted in a better clinical benefit for diabetes therapies. The same advantage was observed when MSC-EVs were administered intratracheally (IT) in an experimental model of hyperoxia-induced bronchopulmonary dysplasia (BPD) in rat pups, showing a significant therapeutic effect of the non-systemic EV injection in combination with a more feasible clinical procedure [23,24].

In this context, the development of labeling strategies able to monitor and quantify EV localization is mandatory. The approaches of the labeling developed can be categorized based on the tags used, the labeling site, and the EV modification, in two main groups: (i) membrane/surface EV labeling by the integration of the lipophilic near-infrared dyes (e.g., DiR) (ii) genetic modification-dependent labeling through the introduction of tagged fluorescent-fusion proteins in the EV-originating cells [25]. Conversely, the MSC-EVs biodistribution is challenging, due to the critical limits in imaging EV dynamics, based on their complex physical properties as well as technical labeling limitations [26].

In our work, we investigated, using fluorescence imaging strategies, the in vivo pharmacokinetic distribution patterns of DiR-labeled MSC-EVs through IV, IT and intranasal (IN) administration into healthy mice, to verify their safety and to distinguish the most efficient delivery system based on the chosen target tissue. We performed a systematic comparison of the three administration routes, evaluating the whole-body mice fluorescent signal at different times from EV administration. The ex vivo organ’s fluorescent signal was also evaluated after 24 h from injection and compared to the signal from homogenized tissues. Direct tracing in vivo of DiR-labeled EVs was performed to identify the potential MSC-EVs target organs.

## 2. Materials and Methods

### 2.1. Extracellular Vesicles

Umbilical cord MSC-EVs (MSC-EVs) were kindly provided by the Cell Factory BVBA (Esperite NV, Niel, Belgium). The MSC and EV production process followed the good-manufacturing-practice (GMP) standards (ICH Q7), based on the ISO9000 quality system. For details related to source material, MSC and EV production platform and quality control, refer to Porzionato et al., 2019 [23]. Briefly, EV were isolated by concentration using tangential-flow filtration and by applying filters with a cut-off of 100 KDa. This approach was instrumental in reducing cell-culture contaminants, with a high rate of EV recovery [23].

### 2.2. Tunable Resistive Pulse Sensing (tRPS)

Particle concentration and size distribution were analyzed using tunable-resistive-pulse-sensing (TRPS) technology with the qNano instrument (Izon Science, Christchurch, New Zealand). NP100 or NP150 membranes were used for the analysis. The concentration of particles was standardized using a CPC100 calibration solution diluted 1:10,000 (110 nm mean carboxylate polystyrene beads; stock concentration 1.00 × 10^12^) [14].

### 2.3. Transmission Electron Microscopy

MSC-EVs were fixed with 4% PFA for 5 min, and one drop (2.00 × 10^9^ particles) of EVs was placed on a 400-mesh holey-film grid for 10 min. After washing with PBS, the MSC-EVs were stained with 1% uranyl acetate for 2 min. The sample was then washed with PBS and finally observed with a Tecnai G2 (FEI; Thermo Fisher Scientific, Waltham, MA, USA) transmission electron microscope operating at 100 kV. Images were captured with a Veleta (Olympus Soft Imaging System; Münster, Germany) digital camera.

### 2.4. Immunophenotyping of MSC-EVs

MSC-EVs were characterized by flow cytometry using the MacsPlex Exosome Kit (Miltenyi Biotec, Bergisch Gladbach, Germany), following manufacturer’s instructions. Briefly, 1.00 × 10^9^ particles of MSC-EVs were loaded in a 1.5 mL tube and diluted to 120 µL with MACSPlex buffer. Then, a mix of 15 µL of exosome capture beads and 15 µL of antibody were added to each tube and incubated for 1 h at RT in agitation. For blank control, only MACSPlex buffer was used. Then, 500 µL of MACSPlex buffer was added to each tube and the EVs were centrifuged at 3000× *g* for 5 min. Finally, 500 µL of the supernatant was discarded and the samples were resuspended and transferred to a flow-cytometry tube. Samples were analyzed using a Cytoflex (Beckman Coulter, Brea, CA, USA) flow cytometer.

For the Western blot analysis, MSCs and their released EVs were lysed in RIPA buffer (50 mM Tris, pH 8.0, 150 mM sodium chloride, 1.0% NP-40, 0.5% sodium deoxycholate, 0.1% SDS (sodium dodecyl sulfate)) plus a protease inhibitor cocktail (Roche, Basle, Switzerland) and incubated with Laemmli buffer with β-mercaptoethanol for 5 min at 95 °C, for complete protein denaturation. Then, 5 µg of protein were loaded and resolved in an SDS-polyacrylamide 4–12% gel at 160 V and blotted using semi-dry transfer for 7 min at 25 V to polyvinylidene difluoride membranes (PVDF) (GE Healthcare Life Science, Milan, Italy). Membranes were blocked with 5% BSA in TBS-Tween for 1 h at room temperature and then incubated with the following primary antibodies overnight: anti-TSG-101 antibody 1:1000 (ab-125011); anti-CD81 antibody 1:500 (ab-109201) anti-CD9 antibody 1:1000 (ab-92726) (all from Abcam, Cambridge, UK) and anti-calnexin 1:1000, (sc-23954; Santa Cruz biotechnology, Dallas, TX, USA), and diluted in 1% BSA in TBS-Tween. After washing, the membranes were incubated with the secondary antibodies: goat anti-rabbit HRP 1:5000 (65-6120; Thermofisher, Waltham, MA, USA) or goat anti-mouse HRP 1:5000 (sc-2005, Santa Cruz biotechnology) for 1h at room temperature. Bands were then evidenced using ECL Plus Western blotting system (32134, Thermofisher).

### 2.5. Labeling and Cellular Uptake of MSC-EVs

The uptake of the EVs was evaluated in the human monocyte cell line (THP-1 cells, ATCC) and human endothelial cells (HMEC cells, ATCC). The THP-1 cells were cultured in RPMI-1640 medium, supplemented with 10% FBS, while the HMEC cells were cultured in EndoGRO-LS complete culture medium (Merck Millipore, Burlington, MA, USA). MSC-EVs were labeled with DiR (2.5 µM per 1.00 × 10^10^ particles) and incubated for 1 h in PBS (DiR-EVs). DiR dye (2.5 μM) was diluted in PBS alone and used as control (DiR-PBS). The free, unbound dye was eliminated from the preparations, by applying 3 serial centrifugations at 4 °C at 2000× *g*, using the 100 kDa Amicon filters (Merck Millipore). The EV-DiR and PBS-DiR were then collected from the filter (around 150 μL). The same staining protocol was used for the in vivo experiments. The THP-1 and HMEC cell lines were seeded in 8-well chamber slides (2.00 × 10^4^ cells/well and 1.00 × 10^4^ cells/well, respectively) and incubated for 24 h. After that, DiR-PBS or DiR-EVs (1.00 × 10^9^ particles/well) were added and incubated for 24 h. Then, cells were fixed in PFA 4% for 10 min and washed and stained with phalloidin-FITC for 20 min. After washing, Vectashield with DAPI was added. Images were taken using a Leica microscope at 40×.

### 2.6. In Vivo Administration

All in vivo experiments involved 6- to 8-week-old BALB/c male mice (Charles River, Wilmington, MA, USA), which were housed in our specific-pathogen-free (SPF) animal facility. Procedures involving animals and their care were in conformity with institutional guidelines that comply with national and international laws and policies (D.L. 26/2014 and subsequent implemented circulars). During in vivo experiments, animals (*n* = 48) in all experimental groups were examined for a decrease in physical activity and other signs of illness.

Intravenous administration. To evaluate the biodistribution of systemically administered DiR-labeled MSC-EVs, the mice were injected intravenously (IV) with 2.00 × 10^10^ DiR-labeled MSC-EVs (100 µL/mouse).

Intratracheal administration. The mice received the same dose of DiR-labeled MSC-EVs in 30 µL/mouse of PBS by intratracheal instillation under isoflurane anesthesia.

Intranasal administration. The anesthetized mice were instilled intranasally (I.N.) with 2.00 × 10^10^ of DiR-labeled MSC-EVs (25 µL/mouse).

### 2.7. IVIS Imaging of In-Vivo-Administered MSC-EVs

DiR-labeled MSC-EVs were injected in BALB/c mice of 6–8 weeks of age via intravenous (IV), intratracheal (IT) or intranasal (IN) administration and monitored over time using an IVIS^®^ Imaging System LUMINA II (PerkinElmer, Waltham, MA, USA). Imaging was performed after injection, at 3 and 24 h after EV administration. After 24 h, the mice were sacrificed, the organs were resected, and images of the organs were captured. Ventral and dorsal images were obtained for each animal and quantified through the region of interest (ROI). We measured the fluorescence as radiance (p/s/cm^2^/sr) in the live animals and excised organs, using the Living Image 3.2 software program (PerkinElmer).

### 2.8. Organ Biodistribution of Labeled DiR-EVs

After weighing each organ, 500 μL of homogenization buffer was added in safe seal tubes (Sarstedt, Numbrecht, Germany) and tissue homogenate was obtained (tissue lyser-Qiagen). The homogenization buffer was composed of 250 mM sucrose, 2 mM EDTA, 10 mM Hepes-Tris (pH 7.6) and protease inhibitors (Sigma Aldrich, St. Louis, MO, USA). After centrifugation at 6000× *g*, the supernatant was recovered, and 150 μL of each organ homogenate was placed in a 96-well black plate with a glass bottom (Corning) and fluorescence was quantified using the Ensight fluorimeter (PerkinElmer, excitation 745 nm and emission 770 nm). The fluorescence intensity of each organ was normalized to the respective organ weight. Furthermore, the DiR fluorescent background of the DiR-PBS of each organ (control) was subtracted from the signal intensity of the DiR-EVs-treated organs (normalized fluorescent intensity, NFI).

### 2.9. Immunofluorescence Staining

OCT- (Biosigma) embedded tissue slices (8 μm) were stained as follows. Tissue slices were permeabilized with Triton X-100 0.1% in PBS for 20 min and blocked with 10% horse serum for 15 min. Incubation with the primary antibody (rat anti-mouse CD31, clone RM5200-invitrogen, MA, USA) was performed overnight at 4 °C. Alexa fluor-488 secondary antibody (A11006-Life technologies, Monza, Italy) was added to the slides for 1 h at 37 °C. The slides were washed and mounted with Fluoroshield™ with DAPI (F6057-Sigma Aldrich). Pictures were taken using Zeiss LSM800 Airyscan confocal microscopy (Jena, Germany) with 63× magnification.

### 2.10. Statistics

All data were processed, and their statistical significance was determined using GraphPad software (version 8.0.2). Data are expressed as mean ± standard deviation (SD). Comparisons of categorical variables were carried out using one-way ANOVA followed by Bonferroni’s multiple-comparison-test *p* values: * *p* < 0.05; ** *p* < 0.01; *** *p*< 0.001; and **** *p* < 0.0001.

## 3. Results

### 3.1. Characterization of MSC-EVs

The analysis of the GMP-grade MSC-EVs by tunable resistive pulse sensing (tRPS) demonstrated a homogeneous population with particle sizes mostly below 150 nm (Figure 1A). Transmission electron microscopy (TEM) confirmed the EV heterogeneity, with particle sizes ranging from 30 to 250 nm (Figure 1B). The tRPS and TEM analysis confirmed, therefore, that almost all the nanoparticles used in the present work can be referred to as “small EVs”, according to the MISEV2018 criteria [14]. MSCs and their released EVs expressed the exosome surface markers CD9 and CD81 and the intravesicular marker TSG101, as detected by Western blot (Figure 1C). On the other hand, the MSC-EVs were negative for the endoplasmic-reticulum protein, calnexin (CNX), which was selectively expressed by MSCs (Figure 1C). The presence of different surface proteins was also analyzed by flow cytometry, using a bead-coupled assay. MSC-EVs were confirmed to be positive for the classical tetraspanin markers CD9, CD63 and CD81. Regarding the adhesion molecules, the MSC-EVs expressed relatively high levels of CD29 and the angiogenic proteoglycan MCSP, while they did not express immunological markers (Figure 1D). Finally, the vesicles’ surface protein reflected those of their parental cells, as they were positive for the CD105, CD44 and SSEA4 markers, while negative for CD45 and HLA-DR. Of note, the absence of HLA-DR would suggest a low immunogenicity (Figure 1D). The MSC-EVs labeled with DiR maintained the capability of being taken up by different target cells (e.g., human endothelial cells and monocytic cells) (Appendix A), confirming no alteration on their surface. 

### 3.2. In Vivo Biodistribution of DiR-Labeled MSC-EVs in BALB/c Mice

To study the biodistribution of the MSC-EVs, we evaluated three different administration routes: intravenous (IV), intratracheal (IT) and intranasal (IN) administration of DiR-labeled MSC-EVs (DiR-EVs) in healthy BALB/c mice. Control mice were injected with DiR-labeled PBS (DiR-PBS). We tracked the DiR-EVs at different time points from administration in live animals (after injection, at 3 and 24 h) using the IVIS^®^ spectrum scanner (Figure 2A). Whole-body analysis demonstrated that in IV-injected mice the accumulation of MSC-EVs was mainly in the abdominal region of the body. On the other hand, in IT-injected mice, the signal was mainly localized at the level of the chest; it was strongly visible 3 h after the EV administration and was maintained at 24 h (Figure 2A). Finally, with the IN injection, the fluorescent signal detected was mainly located in the cerebral region, at both 3 and 24 h, as observed in mice placed in posterior position (Figure 2A). Quantification of fluorescence intensity in regions of interest (ROI) was expressed as the average radiance ± standard deviation (SD) and is depicted in Figure 2B. No significant differences were observed in the intensity of the fluorescent signal among the different administration groups, or at different times after the injection.

### 3.3. Ex Vivo Detection of DiR-Labeled MSC-EVs after IV, IT and IN Injection

To investigate the specific organ distribution of the injected DiR-EVs, the IVIS^®^ spectrum scanner was used to image the harvested organs ex vivo. Images of the organs (brain, gut, heart, liver, lungs, kidneys, and spleen) harvested at 24 h after the EV administration are shown in Figure 3A. The mean radiant efficiency relative to the pixel size of the region of interest for each organ was quantified (Figure 3B). In agreement with the results from in-vivo-imaging experiments, the highest fluorescent signal after IV injection was detected in the liver, followed by the spleen and lungs. Conversely, almost no signal was detected in the kidneys, heart, and brain tissues 24 h after injection (Figure 3B). Similarly, in the IT-injected mice, after 24 h the highest signal was detected in the lungs, while the other organs did not show any significant signal (Figure 3B). Finally, after the IN injection, the signal was detected specifically in the brain, followed by a low detectable signal in the lungs, kidneys and gut (Figure 3B). Results from the DiR-EV injected groups indicated a low overall signal in the blood both 3 and 24 h after the IV, IT and IN injections, which was absent in the DiR-PBS infused mice (Appendix A).

To better characterize the signal derived from the DiR-EVs, the relative fluorescence was assessed also by fluorometric analysis of the homogenized tissue. After homogenization, the highest EV signal, in the IV animals, was detected in the liver and the spleen, followed by the lungs (Figure 3C). In IT-injected mice, fluorescence was mainly observed in the lung tissue, which was consistent with that observed after the IVIS^®^ analyses (Figure 3C). Finally, the IN-injected mice exhibited a low but specific signal at the level of the brain tissue (3266.7 ± 2311.3), followed by the heart and gut (617.5 ± 437.1 and 297.7 ± 213.9, respectively), (Figure 3C). No fluorescent signal was detected in the other organs that were analyzed. 

### 3.4. DiR-Labeled MSC-EV Tissue Detection after IV, IT and IN Administration

Data comparing the relative fluorescent signal obtained from the ex vivo analysis, for the liver, lungs, and brain, for the three administration routes, is shown in Figure 4A. In the liver, the strongest signal was detected after IV injection, compared with IT and IN injection, where the signal was significantly lower. Similarly, in the lungs the highest signal was observed after IT administration, which was significantly higher when compared to IV and IN administration. After IN injection, the highest signal was observed in the brain tissue, but significance was reached only in respect of the IT administration. Finally, immunofluorescence studies were carried out on organs targeted during the treatments, to reveal the localization of labeled EVs after IV, IT and IN injections. Specifically, DiR-EVs were directly identified in the liver, lungs, and brain after IV, IT and IN injection, respectively, as detected by confocal microscopy (Figure 4B). No fluorescent signal was found in the tissues of the DiR-PBS injected mice (Figure 4B).

## 4. Discussion

Extracellular vesicles are regarded as biological messengers, which deliver active molecules among tissues/cells in both physiological and pathological conditions. Functionally, by delivering their molecular content from donor cells into recipient cells, they modulate the recipient cells’ biological responses, supporting their role as therapeutic delivery systems. Biodistribution studies are essential for understanding EVs’ biology and their potential therapeutic applications. To our knowledge, this is the first time a comparative study has been performed, evaluating three different routes for the administration of GMP-grade MSC-EVs in healthy mice, in which a direct comparison between the in situ and ex vivo analysis of organs after IV, IT and IN administration is conducted.

EV biodistribution can be influenced by numerous factors, including their cells of origin and surface markers, as well as the labeling approach. We were able to potentially reduce variability in our in vivo study by controlling these critical variables. For this reason, EVs were isolated from the same GMP-grade MSC sources and administered at an equal dose into BALB/c mice by different routes. Moreover, the classical tetraspanins (e.g., CD9, CD63, CD81) and the specific adhesion molecules (e.g., Chondroitin sulfate proteoglycan-4, also called MCSP, CD44 and CD29) possibly involved in their uptake [27], were identified on the MSC-EV surface.

Focusing on the labeling protocol, numerous works were presented to evaluate the best imaging tools for in vivo and ex vivo detection, monitoring and quantification of EV localization [28]. We performed the EV labeling using a DiR lipophilic dye known for its efficient labeling properties and a deeper tissue penetration with reduced autofluorescence, together with no evidence of induction of EV chemical changes [29]. The MSC-EVs applied in this study show a heterogenous size distribution of <200 nm, which categorizes them as small EVs [30] and supports their ability to distribute homogeneously in different organs, based on their reduced size [21]. In an attempt to compare different injection routes, we chose an interval of time in which the EV signal at whole-body level was detectable, and which was previously used in other biodistribution studies [31,32,33,34,35,36,37]. Moreover, because we employed a labeling agent with a long half-life (5–100 days), 24 h was used as a reference point. 

The EV dose that we applied was the same as the minimum amount used by [15], showing a specific signal without saturation of the liver reticuloendothelial system. Moreover, the EV concentration was ten times more than the minimum dose proposed by [38], to avoid an excess of free dye in the EV preparation before injection. Concomitantly, a purification step through PBS washing was also performed to eliminate unbound dye and enhance the labeling specificity. In our study, we applied a cross-species treatment by injecting xenotransplant MSC-EVs into BALB/c mice with no evidence of immune reactivity. Our data were supported by numerous mice preclinical studies using human-derived MSC-EVs as described in [39], which showed their capability of directly mediating biological effects without evidence of immune response [40,41]. Similarly, in a previous study, human MSC-EVs injected IV in immunocompetent mice showed the same distribution of mouse-derived EVs, supporting the absence of direct influence of the cross-species treatment in the biodistribution analysis [15].

In our model, DiR-labeled MSC-EVs intravenously administered in healthy mice were most efficiently taken up by the liver and spleen, followed by the lungs. Interestingly, Wiklander et al. [15] in the first report on MSC-EV distribution in mice, also identified the liver as the organ with the highest accumulation of bone-marrow-derived EVs. The same biodistribution was observed also for other EV sources or mice strands after IV injection, as reported by [25,31]. Moreover, the IV-distribution pattern among the examined organs was in line with previous works, evaluating the EV-biodistribution at earlier time points [21]. Interestingly, in this setting, endogenous bioluminescent-labeled cord-blood MSC-EVs, demonstrated the same pattern distribution as that of DiR-labeled EVs at the levels of the liver and lungs [42]. We also observed low levels of MSC-EV uptake by the kidneys and, to a lesser extent, by the heart, colon, and brain in IV mice. Previous biodistribution studies agree with our data, showing the increase in renal uptake of MSC-EVs only after acute kidney damage [43,44]. Furthermore, we demonstrated how the peripheral biodistribution of IV-delivered EVs was largely comparable between the two fluorescent-signal-detection methods used in our study. 

After the systemic injection, the labeled EVs were retained in narrow microcapillaries without reaching uniformly the targeted organs [25]. In contrast, the uptake of labeled EVs increases when a local route of administration is chosen. Indeed, in our study, MSC-EVs administered by IT and IN were efficiently delivered to the lungs and brain, where they accumulated at more than double the concentration with respect to the IV administration, confirming the selective distribution of vesicles based on the administration route chosen. Moreover, only a minimum dispersion in other peripheral organs was observed, supporting the absence of off-target effects and EV retention by specific sites in the body when confined delivery is provided.

The different delivery routes can also generate different mechanisms of action. For instance, the IT and IV injections of bone-marrow MSC-EVs in a bleomycin-induced pulmonary-fibrosis model, were able to activate different pathways in the lung, to reduce the damage [45]. DiR-labeled MSC-EVs were efficiently taken up in vitro by endothelial and monocytic cells [46,47], confirming these cells as possible targets of EVs also in vivo, as well as the absence of any impact of the labeling procedure on the EVs’ surface structure.

Another important result of our study was the detection of EV uptake in the central-nervous-system (CNS) compartments, regardless of the route of administration. EVs can enter the brain directly through the blood–brain barrier (BBB) or the choroid plexus epithelium, a more permeable counterpart of the BBB [48]. Intranasal administration of EVs has been proposed as a promising way to treat CNS diseases. When MSC-EVs were injected by IN in our mouse model, the only organ showing a significant accumulation was the brain, despite the resulting minimum signal. In accordance with this, biodistribution experiments in a non-human primate model resulted in a minimum uptake of EVs, even after intranasal delivery [49]. This may be due to the limitations for drug absorption in the nasal cavity, including low intrinsic permeability (e.g., lipophilic–hydrophilic balance), rapid mucociliary clearance, and active enzymatic degradation [50]. Interestingly, the stronger accumulation of MSC-EVs was demonstrated to occur in the brain only when neuro-inflammatory damage was present [6], suggesting that the recipient species’ pathology could strongly affect EV pharmacokinetics and tissue delivery.

## 5. Conclusions

In the present work, we assessed the in vivo biodistribution of DiR-labeled MSC-EVs using different administration routes: intratracheal, intravenous and intranasal. Analysis of the in vivo-imaging data highlights a selective delivery and permanence of the EV cargo in specific organs, based on their route of administration, after 24 h. These results confirmed that, depending on the administration route, it is possible to deliver MSC-EVs to a target organ, limiting their systemic biodistribution and possible off-target effects. 

In this context, the use of lipophilic dyes has several limitations, including their transfer between membranes and their long half-life, with possible preservation of the fluorescent signal after EV degradation [21]. Moreover, these dyes can affect the physico-chemical characteristics of EVs, resulting in altered EV distribution in favor of lipid-rich organs such as the brain [33]. 

Further data on EV biodistribution in multiple pathological states could better highlight the GMP-grade MSC-EVs’ pharmacokinetics in disease.

## Figures and Tables

**Figure 1 pharmaceutics-15-00548-f001:**
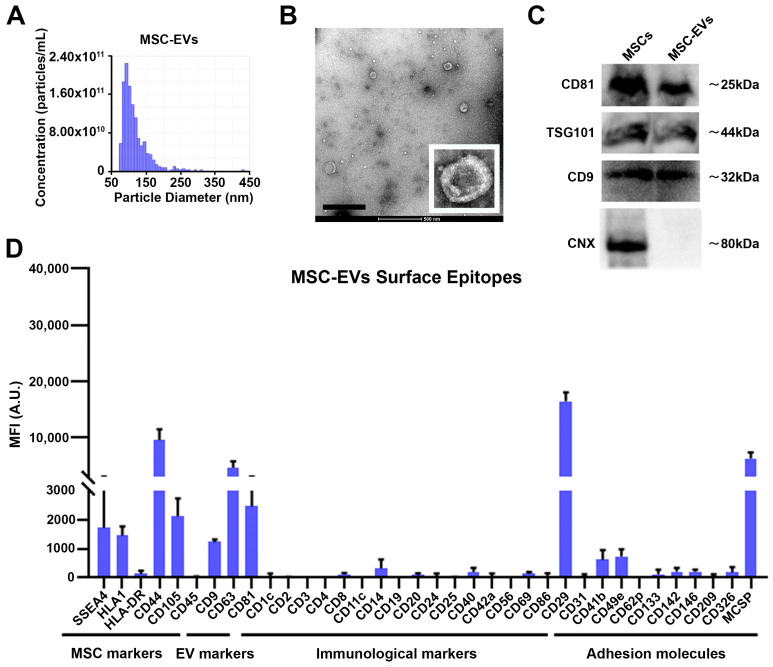
GMP-grade MSC-EV characterization. MSC-EV size distribution was analyzed using tRPS (**A**). MSC-EV morphology was assessed also, using TEM (**B**). Representative images of CD9, CD81, TSG101 and calnexin (CNX) markers analyzed by Western blot in MSCs and MSC-EVs (*n* = 3) (**C**). Finally, EV surface proteins were analyzed by flow cytometry, using a bead-coupled assay (*n* = 3) (**D**). Data are representative of three different independent preparations.

**Figure 2 pharmaceutics-15-00548-f002:**
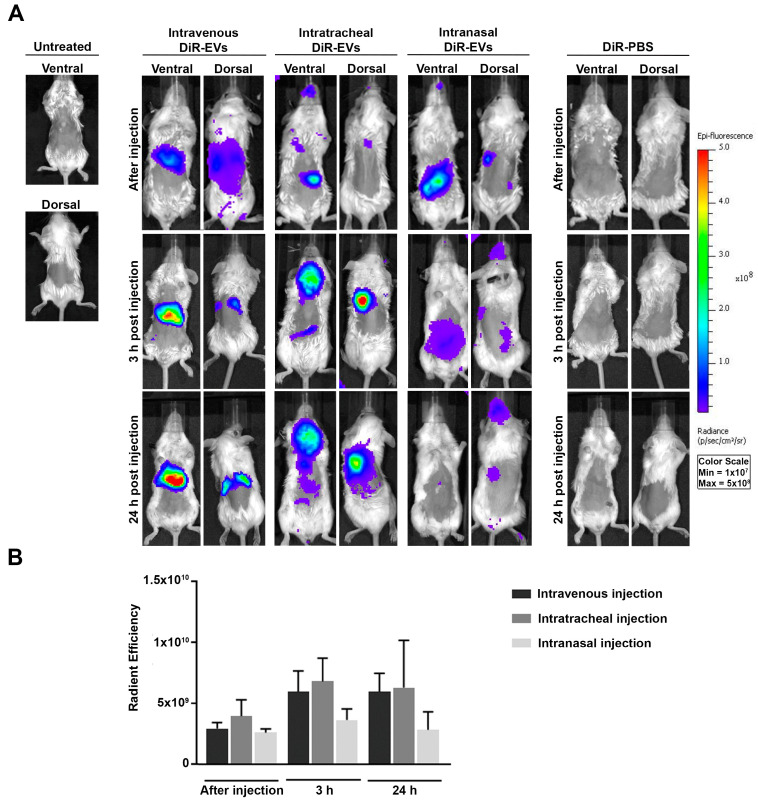
In-vivo DiR-labeled MSC-EV biodistribution. DiR-EVs and DiR-PBS were administered by IV, IT and IN in BALB/c mice, and fluorescence was evaluated in vivo using an IVIS^®^ Imaging System LUMINA II. (**A**) Representative photographs of live animals are presented in supine and posterior position at different times following injection. (**B**) Quantification of fluorescence intensity through the region of interest (ROI) and measured as radiance (p/s/cm^2^/sr). Data are expressed as mean ± SD; (*n* = 24/DiR-EVs and DiR-PBS).

**Figure 3 pharmaceutics-15-00548-f003:**
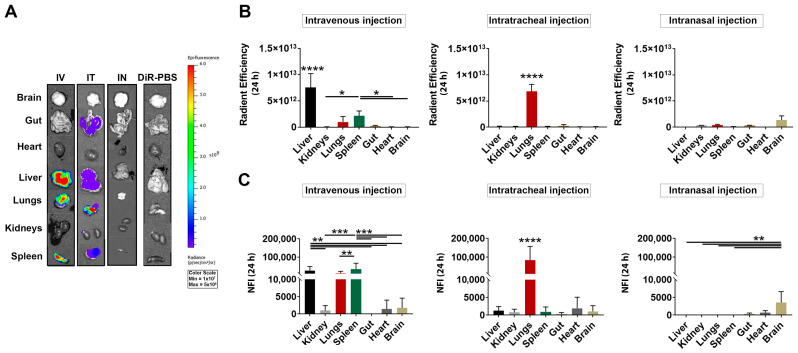
Ex vivo analysis of DiR-labeled MSC-EV biodistribution. Representative image of ex-vivo-organ imaging, using IVIS^®^ (**A**). Quantification of the fluorescence as radiance (p/s/cm^2^/sr) is represented in (**B**) and expressed as average radiance ± SD (*n* = 24/DiR-EVs and DiR-PBS). (**C**) Relative fluorescence in the different organs was assessed using fluorometric analysis after tissue homogenization, and it was calculated as AU/g of tissue. For all the data, subtraction of the DiR-PBS signal was performed (normalized fluorescent intensity, NFI). * *p* < 0.05; ** *p* < 0.01; *** *p* < 0.001 and **** *p* < 0.0001 after one-way ANOVA followed by Bonferroni’s multiple comparison. For **** *p* < 0.0001 against all the other groups.

**Figure 4 pharmaceutics-15-00548-f004:**
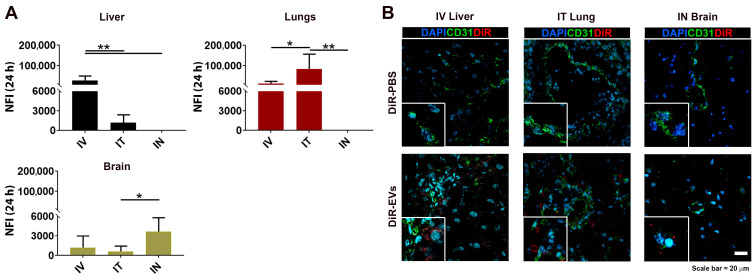
Fluorescent-signal detection in targeted organs. Fluorescence quantification of organ homogenates, where the signal was the highest, is depicted in (**A**); (*n* = 24/DiR-EVs and DiR-PBS) data are expressed as mean ± SD, among the IV, IT and IN injection route. *p*-values: * *p* < 0.05, ** *p* < 0.01 after one-way ANOVA followed by Bonferroni’s multiple comparison. (**B**) Representative images of DiR fluorescent signal and CD31 staining detected on tissue sections of selective targeted organs.

## Data Availability

Not applicable.

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
