# Peer review of "Biodistribution of Intratracheal, Intranasal, and Intravenous Injections of Human Mesenchymal Stromal Cell-Derived Extracellular Vesicles in a Mouse Model for Drug Delivery Studies"

_pharmaceutics, 2023, doi:10.3390/pharmaceutics15020548_

Round 1

Reviewer 1 Report

In their study “Biodistribution of intratracheal, intranasal, and intravenous injected Human Mesenchymal stromal Cell-Derived Extracellular Vesicles in a mouse model for drug delivery Studies” Tolomeo et al investigate the biodistribution of MSC-derived EVs in mice upon delivery by three distinct routes: intraveneous, intratrachael and intranaseal. MSC-EVs have gained increasing interest as novel EV therapeutics. Studies that address their biodistribution under different conditions will help their clinical translation and are thus of high importance.

Introduction:

·         “Ex-vivo organs fluorescent signal and was also evaluated…” The “and” seems to be redundant here

Material&Methods

·         Information on the culture medium of the RAW 264.7 and SVEC4-10 cells is missing. Please add

·         In chapter 2.6 the authors state that the in vivo biodistribution experiments were performed in Balb/c mice, but in chapter 2.7 they mention that NSG mice were used for the experiments. What is true? If indeed NSG mice were used for the experiments, this would severely affect the translational value and results as immune cells are known as one of the main target cell populations of EVs.

·         Chapter 2.9 line 2: Please change the comma to a full stop.

Results:

·         Fig. 1C: The immunoblots shown here used for confirming the presence of EVs lack the respective cell lysates as controls. Moreover, according to the MISEV guidelines I would ask for at least one negative marker to exclude the contamination of EV preparations with intracellular vesicles or cellular debris. In addition, I am a bit surprised as CD81 has a size of around 25 kDa. However, in the immunoblot shown here a band of around 40 kDa was detected… This makes me questioning the specificity of the signal. Has this antibody been validated or can the authors provide a full blot?

·         Please define the abbreviation MCSP

·         Chapter 3.2: Based on published data showing that the half-life of EVs in the circulation is rather short (Kang et al. 2021), can the authors comment why they chose these two time points?

·         Chapter 3.3, line 7: “on” should be “no”

·         Chapter 3.3: The authors state “Finally, IN injected mice exhibited a low but specific signal only at the level of the brain tissue, while no fluorescent signal was detected in the other organs” However, in Fig. 3C some fluorescent signal is visible in the heart tissue as well and should be mentioned.

·         In chapter 3.3 the authors conclude that after IV administration of DiR-labeled EVs, EVs are uptaken mainly by the liver and spleen. I would add the lungs as well since they exhibit quite a lot of signal, especially compared to the signals of the IN administration.

·         In chapter 3.4 the authors state “After IN injection the highest signal was observed in brain tissue, but no significance was reached”. However, in Fig. 4A there seems to be a significant difference between IN and IT?

·         From the IF images in Fig. 4B the authors conclude that the DiR+ EVs are mainly taken up by endothelial cells. However, I cannot detect any co-localization of the DiR signal with the endothelial cell marker CD31. Therefore, I cannot support this conclusion based on the shown data. To support their claim, the authors present in vitro uptake experiments with SVEC cells. Since these images have been taken with a regular fluorescence microscope, it cannot be excluded that the EVs simply adhere to the cell surface without being taken up. If the authors want to confirm EV uptake into (endothelial and immune) cells, they should analyze their samples with a confocal microscope and show co-localization of the respective markers in their in vivo/in vitro images.

·         Fig. S2: The information about the size of the scale bar is missing.

 Discussion:

·         The authors should discuss in how far the use of xenotransplanted human MSC-EVs into mice could have influenced their results

·         I am missing a comparison of the results with the first and major study on EV biodistribution in mice (Wiklander et al. 2015)

·         Discussion, line 26ff: The whole paragraph is almost completely copied from Lázaro-Ibáñez et al. 2021 without even providing the corresponding reference. Please be very careful with such text copying in the future as it could be regarded as borderline plagiarism... Please rephrase!

Author Response

Regards

Reviewer 2 Report

The manuscript by Tolomeo et describes the biodistribution of extracellular vesicles isolated from mesenchymal stromal cells using different administration routes.

Several manuscripts have devoted attention to this particular issue and therefore the novelty is rather limited. Notwithstanding, the side-by-side comparison is useful as a reference for other Researchers. 

Introduction:

1. “.” missing after “…wide range of diseases”

2. “In” missing before “respect to their cells…”

3. Tumorigenic: tumorogenicity was also not evident when MSC were used. 

4. When reference 10 is cited the authors should provide examples.

5. “Ex-vivo organs fluorescent signal and was”: remove “and”

Materials and methods:

1. Provide a brief description of the methods used to isolate the EVs rather than referring to a reference. 

2. Raw concentration should be replaced by “stock concentration”.

3. TEM: mention the number of EVs used.

4. FACS: authors should run EVs without labelling rather than MACSPlex buffer as blank control.

5. Explain better what is “stained free of EVs PBS”

6. Mention the media used for Raw 264.7 and SVEC cells

7. In some places the authors refer to Balb/c mice whereas in other places NGS. Clarify.

8. “24 h after cell injection” should be “after EV injection” I believe.

Results:

1. Provide at least two independent runs for the blot. 

2. In the case of IN injection the signal is not just in the brain and kidneys but specially at 3 h it seems to be distributed in the ventral part. Discuss also why the overall signal is so low.

3. “Almot on signal” should be “almost no signal”.

4. Figure 4A: Y-axis legend is cropped.

5. In Figure 4A the authors say there’s no statistical difference in IN but there’s a “*” between IT and IN…

6. The data presumably showing uptake by ECs and macrophages is not clearly shown. Based on the images that conclusion is not supported by the data.

Discussion:

1. The authors refer to many different citations but the relation to the results is not clearly explained. 

2. The authors should discuss how this results compared with other studies using similar or different doses.

Author Response

Regards

Reviewer 3 Report

In the manuscript by Tolomeo et al, Biodistribution of intratracheal, intranasal, and intravenous injected Human Mesenchymal stromal Cell-Derived Extracellular Vesicles in a mouse model for drug delivery studies. In this manuscript, authors have investigated the distribution profile of MSC-EVs via different routes and concluded that distribution to the organs depends on route of administration. The following comments and/or suggestions should be addressed before publication.

Introduction: Very short, need to elaborate more with previous studies, as there have been many studies where distribution profiles have been performed.

In labelling: How did author confirm that the labelling has been performed on EVs? How did authors confirm the removal of free dye?

-As it has been reported that these EVs have very short half-life, did authors try to find out the half-life of these EVs administered via different route? This would definitely help in their translation into clinical practice as drug delivery system.

Author Response

Regards

Round 2

Reviewer 2 Report

The manuscript has improved substantially. There are some minor issues that need the author´s attention.

1. The authors should provide a reference for the transmigration of EVs. This is a controversial issue and therefore proper references should be provided. Ideally, a critical assessment of those studies should be included.

2. As previously mentioned, there’s no conclusive evidence supporting the tumor-forming capacity of MSCs. The reference provided here is a review article and within that review, the authors also cite another review. Finally, if you look at that review, there is indeed a case report of tumor formation in a patient that received MSCs but that’s only one case. I would explain this point quite carefully. Given the hundreds of persons treated with MSC in clinical trials, I believe the evidence, at least during the time frame considered, does not support the tumor-forming potential. Bear in mind that 1 in 3 persons will develop a tumor throughout their lifetime and this should be taken into consideration when analyzing those findings.

3. “.” missing after references 16-18.

4. Capitalize Thp-1 cells in 2.5 from materials and methods. Also, add “were cultured” before EndoGRO-LS.

5. “Fair above background” means exactly what? It would be wise to replace the statements in this paragraph with quantitative values…

6. In line 403, corroborate should be replaced by “are in line” or “support our results” or “are in accordance with”

7. In lines 419-422 I suggest to re-phrase the sentence. It seems that the cells are uptake by monocytes and ECs but those were the only cells the authors tried. I believe if other cell types are used the results could be similar (maybe the kinetics would change…)

Author Response

We thank the Reviewer for the careful revision. Here point by point answers to the reviewers' questions. 

  1. The authors should provide a reference for the transmigration of EVs. This is a controversial issue and therefore proper references should be provided. Ideally, a critical assessment of those studies should be included.

R1.  We agree with the reviewer about the importance of addressing EVs transmigration, and the corresponding references associated. We added two new references (Ref. 3 and 4), describing the process (See page 1 line 67).

  1. As previously mentioned, there’s no conclusive evidence supporting the tumor-forming capacity of MSCs. The reference provided here is a review article and within that review, the authors also cite another review. Finally, if you look at that review, there is indeed a case report of tumor formation in a patient that received MSCs but that’s only one case. I would explain this point quite carefully. Given the hundreds of persons treated with MSC in clinical trials, I believe the evidence, at least during the time frame considered, does not support the tumor-forming potential. Bear in mind that 1 in 3 persons will develop a tumor throughout their lifetime and this should be taken into consideration when analyzing those findings. Comment in the main.

R2. This is a misunderstanding between the authors and the reviewer, no strong indication of MSC tumorigenic profile is present in the literature. We removed the phrase to be clearer. See page 4 line 175.

  1. Minor comments:

- “.” missing after references 16-18.

- Capitalize Thp-1 cells in 2.5 from materials and methods. Also, add “were cultured” before EndoGRO-LS.

- “Fair above background” means exactly what? It would be wise to replace the statements in this paragraph with quantitative values.

-  In line 403, corroborate should be replaced by “are in line” or “support our results” or “are in accordance with”

- In lines 419-422 I suggest to re-phrase the sentence. It seems that the cells are uptake by monocytes and ECs but those were the only cells the authors tried. I believe if other cell types are used the results could be similar (maybe the kinetics would change…).

R3. We added all the information requested and corrected the typo errors identified along the text.